# Differential Role of Active Compounds in Mitophagy and Related Neurodegenerative Diseases

**DOI:** 10.3390/toxins15030202

**Published:** 2023-03-06

**Authors:** Mark Makarov, Eduard Korkotian

**Affiliations:** Department of Brain Sciences, The Weizmann Institute of Science, Rehovot 7630031, Israel

**Keywords:** mitophagy, autophagy, neurodegeneration, spermidine, urolithin, resveratrol, haloperidol, T-2 toxin, niclosamide, sevoflurane, quercetin

## Abstract

Neurodegenerative diseases, such as Alzheimer’s disease or Parkinson’s disease, significantly reduce the quality of life of patients and eventually result in complete maladjustment. Disruption of the synapses leads to a deterioration in the communication of nerve cells and decreased plasticity, which is associated with a loss of cognitive functions and neurodegeneration. Maintaining proper synaptic activity depends on the qualitative composition of mitochondria, because synaptic processes require sufficient energy supply and fine calcium regulation. The maintenance of the qualitative composition of mitochondria occurs due to mitophagy. The regulation of mitophagy is usually based on several internal mechanisms, as well as on signals and substances coming from outside the cell. These substances may directly or indirectly enhance or weaken mitophagy. In this review, we have considered the role of some compounds in process of mitophagy and neurodegeneration. Some of them have a beneficial effect on the functions of mitochondria and enhance mitophagy, showing promise as novel drugs for the treatment of neurodegenerative pathologies, while others contribute to a decrease in mitophagy.

## 1. Introduction

For adequate interaction with the environment, adaptive behavior in society, and the realization of creative potential, every person needs a properly functioning nervous system (and the brain in particular). The elementary morphofunctional unit of the nervous system is the synapse; the formation of memory is associated with synaptic events. Morphological changes in the synapse correspond to functional requirements; this phenomenon is called neuroplasticity. This process requires careful control by the intracellular apparatus and enhanced energy supply. A normally functioning neuron and synapse are possible only in the presence of healthy mitochondria. In the synapse, mitochondria play the role not only of an energy station, but also as a calcium (Ca^2+^) regulator and an inducer of apoptosis [1,2].

Mitochondrial damage can lead to impaired adenosine triphosphate (ATP) production, an increase in oxidative stress products, and an increase in the level of Ca^2+^ in the cytosol. These undesirable consequences can be prevented through the processes of mitophagy. Mitophagy is a specific type of autophagy in which the maintenance of cellular structure and function is carried out via elimination of unhealthy mitochondria [3,4]. The term mitophagy first appeared in 2005; since then, the popularity of this topic has grown exponentially [5,6]. Due to the role played by mitochondria in the cell, the topic of mitophagy has attracted researchers in anti-aging medicine [7,8], cardiology [9,10], endocrinology [11,12,13], nephrology [14,15], and neuroscience [16,17,18,19].

Mitochondrial damage can occur by various agents associated with drug intervention, such as during anesthesia by sevoflurane [20,21,22,23], when using anticancer drugs (Paclitaxel) [24] or antipsychotic drugs (haloperidol, clozapine) [25,26], after ischemic brain damage [27,28], and in neurodegenerative diseases [18,19]. In all these cases, the mitophagy process is disturbed, and therefore it should be a target for therapeutic intervention in order to enhance the effectiveness of a rehabilitation or preventive program. This circumstance promotes an investigation of compounds that may accelerate mitophagy.

In this review, we have collected the latest data on the study of the effect of different molecules on mitophagy and neurodegenerative processes. We will take a detailed look at exactly how damage to mitochondria contributes to neurodegeneration and how some compounds can influence this through mitophagy. Not being able to cover all possible substances that affect mitophagy, we focused on such compounds that: (1) humans have a risk of exposure to (harmful substances, food components, drugs), (2) are the most studied and highly used ones, and (3) have qualitatively diverse effects on mitochondria. One group contains substances with the prospect of becoming drugs aimed at enhancing mitophagy for therapeutic or prophylactic purposes. The other group of substances refers to drugs that are already being used in medicine, but have side effects, including disruption of the mitochondria and the normal process of their elimination. We hope that both researchers of mitophagy and clinicians will find this review useful for their practice.

## 2. Mitochondria in the Central Nervous System

### 2.1. Role of Mitochondria in the Synapse

Mitochondria are localized in all synaptic compartments: in the presynaptic terminal, at the base of the dendritic spine, in the astrocytic body and processes, and in microglial cells (when considered a part of the synapse). In the presynaptic terminal, mitochondria provide energy (ATP) for the formation of synaptic vesicles, the release of neurotransmitters into the synaptic cleft, and reuptake of the mediator. In dendrites, mitochondria are located at the base of the most functionally loaded dendritic spines, where they are responsible for providing energy for the main stages of synaptic plasticity: phosphorylation, externalization and synaptic recruitment of receptors, and changes in the structure of the dendritic spine. The quality of the mitochondrial state at the base of the dendritic spine, including membrane potential, also determines Ca^2+^ homeostasis, its global and local gradients [1,2,29]. In astrocytic processes, mitochondria support the operation of glutamate transporter-1 (GLT-1), excitatory amino acid transporter (EAAT) 1 and 2, as well as many other functions [30,31,32].

If the quality of the mitochondria is reduced, then the energy supply of all processes will be disrupted. For example, astrocytic mitochondria dysfunction induced with a subsequent decrease in the energy supply for the uptake of glutamate from the synaptic cleft can lead to the process of glutamate excitotoxicity [30]. In the dendritic spine, disruption of the mitochondria can lead to a decrease in the ability to maintain the function of the spine. This violation may be caused both by a decrease in energy supply and by a deficiency in the local Ca^2+^ signaling [33].

Damaged mitochondria contribute to the production of reactive oxygen species that disrupt the structure of macromolecules such as DNA, lipids, and proteins, which ultimately leads to necrosis and/or apoptotic death of the nerve cell [34,35].

The number of processes that depend on mitochondria in synapse is equal to all currently known molecular mechanisms in this area and requires a separate review. We have previously reported in more detail on the role of mitochondria in this area [1]. Thus, a functional deficiency of synaptic mitochondria leads to negative changes in this area (Figure 1). That is why mitochondria and the processes of maintaining their quality have become such an important target for therapeutic intervention.

### 2.2. Role of Mitophagy in Neurodegeneration

To understand how various compounds can determine the course of neurodegenerative pathologies through mitophagy, it is necessary to understand the role of mitophagy in neurodegeneration. If the violation of mitophagy occurs because of pathological processes of neurodegenerative pathology, then these compounds are cofactors that can change the course of pathology through mitophagy. If the disruption of mitophagy is primary in relation to neurodegenerative processes, then these compounds are toxins that may receive a status of etiological factors. This provision will significantly change the understanding of neurodegenerative pathologies and the role of surrounding compounds.

Alzheimer’s disease is one of the most common neurodegenerative diseases characterized by a progressive impairment in cognitive functions and episodic memory, changes in behavior and personality, followed by social and everyday maladaptation. Genetic variants of AD occur only in 1–2% of cases and are associated with mutations in the APP, PS1, and PS2 genes [36,37]. At the tissue and cellular level, the disease is characterized by extracellular deposits of Amyloid beta (Aβ) plaques and intraneuronal accumulation of pTau [37,38]. Mitochondria are involved in the pathogenesis of AD, and their damage contributes to the progression of AD. On the one hand, the work of mitochondria is disrupted under the influence of the pTau protein and Aβ. Aβ and pTau contribute to damage to mitochondrial DNA (mtDNA), the respiratory chain, a decrease in cytochrome oxidase activity, damage to proteins responsible for the transportation of mitochondria along the neuron, and proteins responsible for normal fission–fusion of mitochondria. As a result, the mitochondrion ceases to provide the cell with a sufficient supply of ATP, the production of reactive oxygen species (ROS) increases, the balance of fission and fusion of mitochondria is disturbed in favor of excessive fission, and the process of elimination of damaged mitochondria in the soma is disturbed. Disruption of the mitochondrial permeability transition pore (mPTP) through the effect of Aβ on cyclophilin D (CypD) leads to severe mitochondrial stress and apoptosis [36,39]. On the other hand, disruption of mitochondrial function promotes an increase in the production of pTau and Aß. For example, in AD models, the disruption of complexes I and III with rotenone and antimycin A increased the level of ROS, leading to the accumulation of Aβ [40]. In addition, mitochondrial dysfunction leads to hyperphosphorylation of Tau [38,41]. The involvement of mitochondria in AD pathogenesis is multifaceted and complex. Stimulation of mitophagy in various AD models contributed to the leveling of AD manifestations [42,43,44].

Parkinson’s disease (PD) is characterized by typical motor disturbances, including bradykinesia, tremor, rigidity, and postural unsteadiness, as well as a range of non-motor symptoms. The pathological picture of PD is associated with progressive loss of dopaminergic neurons and aberrant accumulation of α-synuclein (α-syn) in the form of Lewy bodies in the substantia nigra compactus (SNpc). Mitophagy disorders have been found in patients suffering from PD and examined postmortem [17,45]. α-syn can lead to a deficiency of complex I, decreased ATP production and increased ROS levels, membrane depolarization, and the release of cytochrome c into the cytosol, leading to apoptosis. α-syn binds to endoplasmic reticulum (ER) or mitochondria-associated membrane (MAM) membranes, reducing the interaction between mitochondrion and ER, promoting mitochondrial fragmentation and disruption of mitophagy. This fragmentation may also be associated with disruption of the mitochondrial fission factors (Drp1, Opa1) [45]. At the same time, mutations in the genes of proteins involved in mitophagy PARK6 (coding for PINK1) and PARK2 (coding for Parkin) are associated with autosomal recessive PD; however, whether PD is caused by a violation of mitophagy in this case remains to be seen. It should be considered that there are compensatory pathways of mitophagy; therefore, the development of PD may be associated with other functions of the PINK1 and Parkin proteins [45,46].

The main problem of neurodegenerative diseases is a lack of mitophagy, but in some pathologies, excessive mitophagy can be a negative pathophysiological link. Neurological pathologies with excessive mitophagy include stroke [28,47] and multiple sclerosis (MS) [48,49]. The cell needs to maintain a balance between the number of mitochondria and mitochondrial health in order to provide enough energy on the one hand, and on the other hand to maintain other homeostatic parameters.

Thus, mitophagy in neurodegenerative processes is an important link in the self-sustaining feedback loop of neurodegeneration, which is why the search for and study of compounds that contribute to the proper course of mitophagy is so important.

### 2.3. Mechanisms of Mitophagy

Mitophagy is provided by mitochondria-specific mechanisms for the elimination of damaged mitochondria. The purpose of mitophagy is to isolate the cell from damaged mitochondria and to create space for new healthy ones. This process is carried out with the help of phagophore formation and utilization by lysosomal enzymes. There are different signaling pathways that enable the formation of phagophores. Traditionally, these pathways are divided into those that are Parkin-dependent and Parkin-independent. The Parkin-dependent mitophagy signaling pathway initiates the appearance of PTEN-induced putative kinase 1 (PINK1) on the membrane. PINK1 stabilizes on the outer mitochondrial membrane in response to stress, then autophosphorylates, leading to the accumulation of Parkin on the mitochondrial surface. Parkin promotes ubiquitination of the surface of membrane proteins. Ubiquitins promote attachment of mitochondria to the autophagosome membrane via p62 protein and microtubule-associated proteins with 1A/1B-light chain 3 II (LC3II) [50,51]. Appearance and stabilization of PINK1 on the membrane surface is associated with a decrease in mitochondrial membrane potential [52,53]. Another important link in the induction of the PINK1/Parkin pathway of mitophagy may be ataxia-telangiectasia mutated protein kinase (ATM), which is thought to phosphorylate PINK1. Knockdown of the genes responsible for ATM synthesis leads to disruption of mitophagy stimulated by lead (Pb) or spermidine [54,55].

Parkin-independent mitophagy pathways are activated in response to hypoxia and are associated with the stabilization of BNIP3, Nix, and FUNDC1 proteins on the outer mitochondrial membrane. These proteins, in turn, can bind to LC3II, promoting autophagosome organization [16,56].

Dynamin-related protein 1 (Drp1) plays an important role in mitochondrial division. In the region of the mitochondrial fission ring, Drp1 attaches to fission 1 (Fis1), mitochondrial fission factor (Mff), and mitochondrial dynamic 49/51 kDa protein MiD49/51 proteins, contributing to further division processes [57,58]. Drp1 is involved in Bnip3-dependent apoptosis by controlling the permeabilization of the outer mitochondrial membrane [58,59].

The mechanisms of mitophagy are not isolated from the mechanisms of other processes that may be associated with common autophagy pathways, mitochondrial biogenesis processes, and mitochondrial fusion and division processes. All of the above are controlled by higher order signaling pathways. The main molecules of such signaling pathways are mammalian target of rapamycin (mTOR), adenosine monophosphate (AMP), activated protein kinase (AMPK), and transcription factor nuclear factor erythroid 2-related factor 2 (Nrf2). Changes in glucose and ATP levels affect the functioning of mTOR and AMPK. A decrease in the level of ATP and an increase in AMP contribute to the activation of autophagy and mitophagy. AMP-activated protein kinase (AMPK) is a highly conserved sensor of low levels of intracellular ATP that is rapidly activated after almost all mitochondrial stresses, even those that do not disrupt the mitochondrial membrane potential [60,61]. In turn, AMPK activates unc-51-like kinase 1 (ULK1), which phosphorylates various proteins necessary for normal mitophagy, such as Parkin, TANK-binding kinase 1 (TBK1), and Mff, Beclin-1. Phosphorylated TBK1 itself phosphorylates other proteins regulating mitophagy. Beclin-1 is an important protein for autophagosome formation and recruitment of Parkin [62,63,64]. In contrast to AMPK, mTOR is a major inhibitor of autophagy and mitophagy. It is activated in response to various stimuli, including those caused by an increase in the amount of nutrients and insulin levels, performs its functions as part of the mTORC1 complex, and inhibits ULK1 [12,65]. Nrf2 regulates the expression of many enzymes with antioxidant and detoxifying functions. Nrf2 does not have an inhibitory or activating effect on mitophagy, but participates as a mitophagy regulator, stimulating mitochondrial biogenesis factors (Nrf1, PGC-1α) on the one hand, and mitophagy factors (p62, PINK1) on the other hand [66,67,68,69]. The summary of different possible signaling pathways of mitophagy are presented in Figure 2.

## 3. Effect of Different Compounds on Mitophagy

### 3.1. Niclosamide

Niclosamide is a classic anthelmintic drug for the treatment of cestodosis, and is effective for infestations with Taeniarhynchus saginatus (beef tapeworm), Diphyllobothrium latum (broad fish tapeworm), and Hymenolepis nana (dwarf tapeworm). The anticestodic effect of niclosamide is manifested by the inhibition of oxidative phosphorylation and stimulation of adenosine triphosphatase activity in mitochondria [70]. There are other signaling pathways that can be affected by the drug [71], as well as a range of diseases in which the potential of niclosamide can be unleashed. These diseases are both viral (SARS-CoV, Flavivirus, Hepatitis C Virus, Ebola Virus, Human Rhinovirus, Chikungunya Virus, Human Adenovirus, and Epstein–Barr Virus) and non-viral (such as cancers and metabolic disorders) [72]. Niclosamide is a hydrophobic substance with low solubility and bioavailability (Figure 3); therefore, research is underway on chemical modifications of niclosamide to increase bioavailability and therapeutic efficacy [73].

Barini et al. found that niclosamide and its analogues activate PINK1 in cells due to reversible alterations to the mitochondrial membrane potential, including neurons taken from E16.5 mouse embryos, in cells of pathological relevance to PD [74]. Since then, results have appeared on the effect of the drug on the cells of the nervous system during stress induction. Kato, Y., and Sakamoto, K. (2021), in a cell model of amyotrophic lateral sclerosis (ALS), showed that this drug attenuates morphological changes under stress, activates mitophagy via the PINK1-Parkin-ubiquitin pathway, prevents transactive response DNA binding protein 43 kDa (TDP-43) mislocalization, and promotes degradation of TDP-43 aggregates, and therefore may be a candidate drug for ALS treatment [75]. The summary of experimental data regarding the effects of niclosamide and the compounds described below, is presented in Table 1.

### 3.2. Toxin T-2

T-2 toxin is a type A trichothecene mycotoxin produced by various Fusarium species. T-2 toxin may cause poisoning in humans and animals [114,115]. T-2 toxin is freely soluble in organic solvents but has low solubility in water. T-2 toxin is stable in chloroform and methanol for several days and resistant to various environmental factors [116,117]. This toxin can be found in crops of corn, wheat, barley, and rice in the field, during storage or in drinking water—in particular, it was found in a number of provinces in China [118]. T-2 toxin can be easily absorbed through the intestines and penetrate the blood–brain barrier, lung mucosa, and skin system, damaging the skin, kidneys, liver, brain, heart, spleen, hematopoietic system, lymphoid system of the gastrointestinal tract, bone marrow, and reproductive system [78]. Neurotoxicity caused by T-2 toxin is associated with the production of ROS, oxidative stress, mitochondrial dysfunction, cell cycle arrest, DNA damage, and inflammatory responses [76,77]. Neurological symptoms of poisoning can be muscle weakness, depression and ataxia, and anorexia [79]. The above circumstances highlight the epidemiological significance of T-2 toxin.

The study of mitophagy induced by T-2 toxin on rat pituitary GH3 cells revealed a critical role of the Nrf2/PINK1/Parkin pathway [80]. In renal epithelial cells of Parkin-knockout rats, PINK1/Parkin-mediated mitophagy mitigated T-2 toxin-induced nephrotoxicity [81]. Wu, J. et al. succeeded in activating Drp-1-mediated excessive mitophagy in Leydig cell culture in a concentration-dependent manner and this mitophagy led to apoptosis [82].

To determine the therapeutic potential of the toxin against neurodegenerative pathologies, Sun T. et al. (2022) conducted a study on cultured microglial cells. Microglial cells are actively involved in maintaining homeostasis in brain tissue, but can also cause neuroinflammation, contributing to the emergence of neurodegenerative pathologies [76,119]. During the experiments, it was possible to determine that T-2 toxin causes neurotoxic effects, as well as induces a loss of mitochondrial membrane potential in BV2 microglial cells. Exposure to T-2 toxin activates autophagy in the cell, and this autophagy plays a protective role, accompanied by increased expression of Beclin1 and LC3II proteins [78]. Yet, to date, there are no published articles on the role of T-2 toxin in relation to mitophagy in neurons or astrocytes (see Table 1).

### 3.3. Urolithin A

Urolithin A belongs to a class of substances with α-benzocoumarin in base (Figure 4). Urolithins are formed in the gut microbiota from naturally occurring polyphenols such as ellagitannin or ellagic acid [120,121,122]. The bacteria responsible for the conversion of polyphenols to urolithins in the human intestine is still unknown [120,123]. It is assumed that Gordonibacter urolithinfaciens and Gordonibacter pamelaeae are responsible for the synthesis of urolithin in the intestine [56,124]. Foods rich in metabolic precursors of urolithins are pomegranate, strawberries, walnuts, almonds, persimmons, raspberries, black raspberries, peaches, and plums [125,126,127].

Urolithin A activates Nrf2, contributing to the regularization of mitophagy and mitochondrial biogenesis [66,67,68,69]. There is evidence that urolithin A promotes the activation of the AMPK pathway, which prevents apoptosis of cells with damaged mitochondria and directs biochemical processes towards mitophagy [85,87]. In SH-SY5Y neuroblastoma AD model cells with glucose-induced amyloidogenesis, the destruction of the AIP-AhR complex leads to the expression of type 2 transglutaminase (TGM2) and the formation of MAM; TGM2 promotes an increase in mitochondrial Ca^2+^ inflow from the ER, which is accompanied by an increase in expression of amyloid precursor protein (APP) and β-secretase-1 (BACE1), an increase in Aβ, and subsequent degradation of neurons. In studies on these models, Urolithin A prevented the destruction of the AIP-AhR complex, thereby suppressing the above effects. The same result was noted in vivo [44].

In one of the rare studies of the role of mitophagy inducer in glial cells, it was shown that Urolithin A alleviates neuroinflammation and enhances microglial phagocytosis by increasing mitophagy, which leads to a decrease in the accumulation of tau-protein and beta-amyloid [43,83]. In another study on the BV2 microglial cell PD model, it was shown that neuroinflammation is mediated by an increase in ROS and NOD-like receptors (NLR) family pyrin domain containing 3 (NLRP3), and urolithin A, due to its effect on mitophagy, reduced neuroinflammation and provided a neuroprotective effect [84].

There are many studies on the positive effect of urolithin A on various processes, including life expectancy in C. elegans nematodes [128], on the state of the vascular bed in Wistar rats with manifestations of antisclerotic activity [129], and on cognitive processes [Gong, Z. 2019]. In 2018, urolithin A was officially recognized as a safe substance and approved for human studies [130,131]. Additionally, in 2019, for the first time, a study of urolithin A was conducted in the elderly to assess the safety profile [86]. Thus, urolithin A is the most promising compound for the role of a drug for the prevention and treatment of neurodegenerative diseases (see Table 1).

### 3.4. Resveratrol

Resveratrol (trans-3,5,4′-trihydroxystilbene) is a non-flavonoid polyphenolic compound that has two phenolic rings connected to each other by an ethylene bridge (Figure 5). There are two isomeric forms of resveratrol: cis and trans. It is trans-resveratrol that has greater biological activity [132,133,134,135]. Trans-resveratrol is a photosensitive compound and converts to cis-resveratrol under irradiation [136,137]. This compound is found in grapes, peanuts, berries, in the bark of some plants, seeds, nuts, flowers, and in the plant Polygonum cuspidatum, used in Japanese folk medicine [135,138]. Polyphenols accumulate in plants in response to exogenous stress factors such as trauma, fungal infections, or UV exposure [134,139].

It is known that resveratrol reduces oxidative stress by suppressing the formation of reactive oxygen species, derivatives of NADPH oxidase, and the activation of Nrf2 to maintain endogenous antioxidant protection [90]. By activating Nrf2, resveratrol more likely exhibits cyto- and mitoprotective properties than the properties of a pure mitophagial inducer, since in experiments with the induction of mitophagy using cadmium, resveratrol attenuated the overexpression of p62 and PINK1/Parkin, which suppressed mitophagy and ultimately restored mitochondrial homeostasis [89]. In other situations, resveratrol stimulated mitophagy; for example, in an experiment on a model of Alzheimer’s disease, it was demonstrated that resveratrol promotes mitophagy in Aβ-induced PC12 cells, thereby attenuating oxidative damage to neurons caused by Aβ-peptide [88]. In metabolic disorders, as in the experiment with the induction of endothelial dysfunction, resveratrol activated mitophagy through Bnip3 [91] (see Table 1).

### 3.5. Quercetin

Quercetin is a flavonoid compound found in fruits, vegetables, berries, grapes, wine, various seeds, and nuts. Quercetin has a hydrophobic structure, and the presence of a hydroxyl group contributes to its reducing activity and, accordingly, antioxidant properties (Figure 6) [140,141,142]. In addition to antioxidant properties, quercetin is characterized by anti-inflammatory, antibacterial, antiviral, hepatoprotective, and immunomodulatory activity [96,143].

Quercetin affects many signaling pathways in the cell; in applying to mitophagy, it is able to modulate the process by influencing AMPK and Nrf2, contributing to the modulation of the expression and activity of PINK1/Parkin-mediated mitophagy [96,97].

There are many studies on the positive effects of quercetin in models of neurodegenerative pathologies. Among these effects, a decrease in the level of hyperphosphorylated Tau and amyloid plaques in AD models, a decrease in neuroinflammation with a decrease in microglia and astrocyte activation, and a mitoprotective effect with an increase in ATP production and a decrease in ROS levels were found [92,93]. In studies of mitophagy in PD rat models affected by 6-hydroxydopamine (6-OHDA), in vitro quercetin reduced oxidative stress, increased PINK1 and Parkin levels, and reduced α-synuclein expression. In vivo, quercetin mitigated progressive motor disorders, reduced neuronal death, and reduced mitochondrial damage and α-synuclein accumulation, while these effects were eliminated by knockdown of PINK1 or Parkin [94]. In cultured postnatal rat hippocampal microglial cells, quercetin stimulates mitophagy, thereby preventing neuronal damage by inhibiting mtROS-mediated activation of the NLRP3 inflammasome [95] (see Table 1).

### 3.6. Haloperidol

Haloperidol is a typical first-generation antipsychotic drug belonging to the butyrophenone group (Figure 7). Butyrophenones are prescribed to patients with positive symptoms: delusions, mania, psychosis, and schizophrenia. In addition to antagonistic activity against dopamine D2 receptors, butyrophenones also have activity against several other central nervous system receptors, including other dopamine receptor subtypes, serotonin (5-HT) receptors, and α-adrenergic receptors [144]. The neuroleptic and cataleptic action of haloperidol is associated with the blocking of dopamine D2 receptors in the mesolimbic pathway [145,146,147,148], and side effects such as extrapyramidal symptoms are provided by nigrostriatal pathway receptors [149].

With long-term use, haloperidol causes neurotoxic effects, and the most likely cause of this phenomenon is oxidative stress, which can lead to cognitive dysfunction and motor impairments such as haloperidol-induced tardive dyskinesia [25,98,99,100]. It has been shown that, in addition to increasing oxidative stress, haloperidol can reduce the level of autophagy in SH-SY5Y cells and rat neurons, but the mechanism of this phenomenon is not clear [25,26].

The suppression of mitophagy, as it turns out, cannot be considered only a negative property of one or another compound. There is a hypothesis that demyelination in multiple sclerosis occurs due to impaired mitophagy in glial cells [48,150]. Patergnani et al. (2021) showed that in in vivo MS models, demyelination is associated with excessive mitophagy, and inhibition of autophagy and mitophagy by haloperidol and clozapine leads to the restoration of myelin production and axonal myelination. Treatment of mouse models with these drugs contributed to the improvement of motor activity up to the complete elimination of behavioral disorders [49] (see Table 1).

### 3.7. Sevoflurane

Sevoflurane is an ether drug used for inhalation anesthesia (Figure 8). The low solubility facilitates a rapid onset of action, and low blood solubility also promotes rapid recovery from anesthesia. Sevoflurane has a dose-dependent inhibitory effect on the central nervous system, cardiovascular system, and respiratory tract [151]. It has been used in medical practice since 1990 [152]. The use of anesthetics is known to be accompanied by postoperative complications from the nervous system, such as decreased concentration and attention, cognitive decline, and mental and behavioral disorders [21,101,102]. The issue of the consequences of the use of general anesthetics in children is especially acute, since the developing brain is most susceptible to the toxicity of this group of drugs [101,153]. These disturbances may be related not only to the consequences of the drug’s overall effect on the nervous system, but also to the effect of the drug on mitochondrial dynamics. In studies of the effect of the drug on the cells of the nervous system, it was repeatedly confirmed that sevoflurane stimulates the apoptosis pathway, thereby providing a neurodegenerative effect [20,21,22,23].

The use of various mitophagy inducers contributed to a decrease in the toxic effect of sevoflurane. Experiments have shown that the use of methylene blue [22], honokiol [21], and dexmedetomidine [103,154] contributes to the activation of mitophagy factors and mitigation of sevoflurane’s toxic effect. These data are indirectly supported by studies of the protective effect of sevoflurane in the late phase of reperfusion injury of the heart. The protective effect was provided by the inhibition of mitophagy and the reduction of excessive production of Drp1 and Parkin [104,105]. Therefore, it can be said with certainty that some of the causes of the neurotoxic and neurogenerative effect of sevoflurane are related to mitophagy disorders (see Table 1).

### 3.8. Spermidine

Spermidine is a natural polyamine found in wheat germ, soybeans, mushrooms, nuts, vegetables such as fresh green peppers, cauliflower, broccoli, and enzymatic products such as cheese and red wine (Figure 9). The Mediterranean diet is especially rich in spermidine [111,155,156]. Despite the fact that the replenishment of spermidine reserves occurs not only due to external sources, but also due to synthesis in cells of the body and intestinal microbiota, with age, the amount of spermidine in the human body decreases, so a person needs a diet with a sufficient amount of spermidine [111,156].

Spermidine has three positively charged amino groups, which are very important for its different positive effects [157]. At the cellular and molecular levels, spermidine protects proteins from glycation [158,159]; exhibits antioxidant properties [160,161,162]; is involved in a number of biological processes such as cell proliferation, cell cycle, and apoptosis [54,111]; has an anti-inflammatory effect by suppressing NFκB-dependent pro-inflammatory cytokines [163]; and promotes autophagy induction by inhibiting EP300 acetyltransferase, leading to deacetylation of a cytosolic autophagy-associated protein [164]. At the level of organs and the organism, the above effects contribute to the cardioprotective effect [165], reducing the risk of tumor formation [111,112], prevention of neurodegeneration through demyelination [166,167], and increased lifespan in yeast, flies, human cells [168], and mice [165,169].

In addition to the induction of autophagy pathways, spermidine can stimulate mitophagy. As a mitophagy inducer, spermidine activates ataxia-telangiectasia mutated protein kinase ATM and induces mitochondrial membrane depolarization, thereby promoting the PINK1/Parkin-dependent mitophagy pathway [54,107,111]. The nonspecific effect on mitophagy of spermidine is manifested by mTOR inhibition and AMPK activation [112,113].

The positive effects of spermidine on cognitive processes have been well studied. Spermidine improved olfactory memory in models with age-related memory impairment drosophila flies [106]. In a recent study on the effect of spermidine in C. elegans models of AD and PD, spermidine was shown to inhibit memory loss in AD models and improve behavioral performance in PD models [107]. In experiments in mice with accelerated aging SAMP8, spermidine increased the expression of neurotrophic in neurons, reduced memory loss in the object recognition test (ORT), and significantly relieved tension in the open field test (OFT) [108]. Spermidine is recognized as a safe substance for humans. A blind placebo-controlled study was conducted in people at risk of developing Alzheimer’s disease. According to Wirth, M. et al., spermidine improved episodic memory in mnemonic similarity tests (MST) in older adults with subjective cognitive decline [109,110]. Thus, spermidine, as a natural compound, along with urolithin A, has excellent potential for implementation as a drug for the treatment and prevention of neurodegenerative pathologies (see Table 1).

## 4. Discussion

The study of the role of toxins and different compounds in the processes of mitophagy and neurodegeneration is still at an early stage. As we have seen in the example of the above compounds, substances of various sources can influence mitophagy, such as drugs, food components, fungal toxins, and products of the intestinal microflora. We have no clear idea about the effect of air pollutants on mitophagy. So far, little information has been accumulated on inhibitors of mitophagy. However, it should be considered that different molecules that can directly or indirectly affect mitophagy are around us, and their influence can be long-lasting and contribute to a certain state of mitophagy for many years, causing various effects on cells, tissues, and organs.

A very promising direction in the field of the role of different compounds in mitophagy is the study of participation of the intestinal microflora. The human body is completely covered with a layer of microbiota that produces metabolites, and these metabolites can have the properties of inhibitors or inducers of mitophagy. Among the intestinal bacteria that produce mitophagy inducers, we know Gordonibacter urolithinfaciens and Gordonibacter pamelaeae, which metabolize urolithins from ellagitannin and ellagic acid [56,124]. In studies by Li, F. et al. it was found that intestinal microflora metabolite nicotinamide n-oxide promotes mitophagy in microglia, thereby preventing herpes simplex encephalitis [170,171]. In a recent study, it was demonstrated that Streptococcus thermophilus FUA329 can produce urolithins from ellagic acid, which opens the prospect of a new generation of probiotics [172]. It can be assumed that intestinal dysbacteriosis or sterilization can lead to suppression of the production of mitophagy inducers by the intestinal microbiota, and that was shown in experiments on mice with Ochratoxin A-induced hepatitis. Antibiotic-treated mice showed reduced liver mitophagy [173]. Articles about inhibitors of mitophagy produced by the intestinal microflora were not found. Nevertheless, the intestinal microflora synthesizing at least only mitophagy inducers, in the light of recent data, becomes an active participant in maintaining the qualitative composition of mitochondria, and therefore a participant in maintaining the normal functioning of the nervous system and in pathological processes during neurodegeneration (see Figure 10).

The ability of compounds to determine the state of the entire nervous system through mitophagy attracts the search for the study of compounds that affect mitophagy. Urolithin and spermidine are being introduced into medical practice; however, these substances already have a natural origin and are known for their participation in the functions of the human body, which makes it relatively easy to introduce these substances as drugs that affect mitophagy. The study is not limited to these substances only, and the search for mitophagy inducers extends to the toxins of fungi and bacteria, various synthetic substances, and existing drugs. Data for implementation in medical practice must meet the following requirements:The compound must not damage healthy mitochondria, or the damage to healthy mitochondria must be reversible. The stimulation of mitophagy in each of the considered compound occurs for an unknown reason. For example, niclosamide promotes a reversible decrease in the mitochondrial membrane potential, leading to the stabilization of PINK1 on the surface of the outer mitochondrial membrane [74]. However, it is unclear what contributes to the decrease in membrane potential. One may assume that the reversibility of the decrease in the membrane potential indicates the complete restoration of the mitochondria in its structural and functional respect. In our opinion, the preservation of the function of mitochondria that has not been eliminated by mitophagy when exposed to any compound does not yet indicate the safety of the substance. Mitochondria have a fairly robust quality control mechanism. However, over time, mutations can occur in the mitochondria, which are fixed and transmitted to the daughter ones. These changes are not critical for the short-term survival of mitochondria, but accumulate from generation to generation, leading to a gradual decrease in function [174]. We hypothesize that mitophagy inducers may promote additional mitochondrial mutations. During experiments on the induction of mitophagy, the positive effect of increased elimination of damaged mitochondria may outweigh the insignificant negative chronic effects of the inducer, which suggests that the inductor has a positive effect on mitophagy, on cell survival in certain models, on survival, and on the results of laboratory functional tests in animals. However, the life expectancy of laboratory animals is usually incommensurable with the life expectancy of a person. We believe that it would be appropriate to conduct long-term testing of a drug that affects mitophagy, for example, in cells with an infinite number of division cycles to monitor whether the chronic toxic effect of a mitophagy inducer will negatively affect mitochondrial function in the long term.The compound should activate mitophagy only if mitophagy is impaired. An increase in normal mitophagy above physiological range can lead to depletion of the mitochondrial pool in cells and aggravation of neurodegenerative processes due to a decrease in the supply of ATP energy to the cell during drug exposure.A marker of the degree of mitophagy is needed to select the dosage of the drug. In a placebo-controlled study of urolithin A, a thigh muscle biopsy was taken from the subjects and markers associated with the activation of mitochondrial genes under the influence of urolithin A were measured [86]. In clinical practice, this method can be considered barbaric; therefore, methods for determining mitophagy markers directly in plasma are currently being developed [175,176].

## 5. Conclusions

The role of different compounds in the processes of mitophagy and neurodegeneration is determined by the range of disorders of mitochondrial functions in the nervous tissue. Insufficiency of mitophagy and, as a result, a decrease in the ability of mitochondria to produce ATP, maintain Ca^2+^ homeostasis, and increase ROS production, can be corrected with the help of mitophagy inducers, and vice versa, some substances can have a detrimental effect on the mitophagy process or be useful when excessive mitophagy contributes to pathology process. The study of mitophagy and the development of new drugs for the treatment of neurodegenerative pathologies determine a significant contribution to modern neurology.

## Figures and Tables

**Figure 1 toxins-15-00202-f001:**
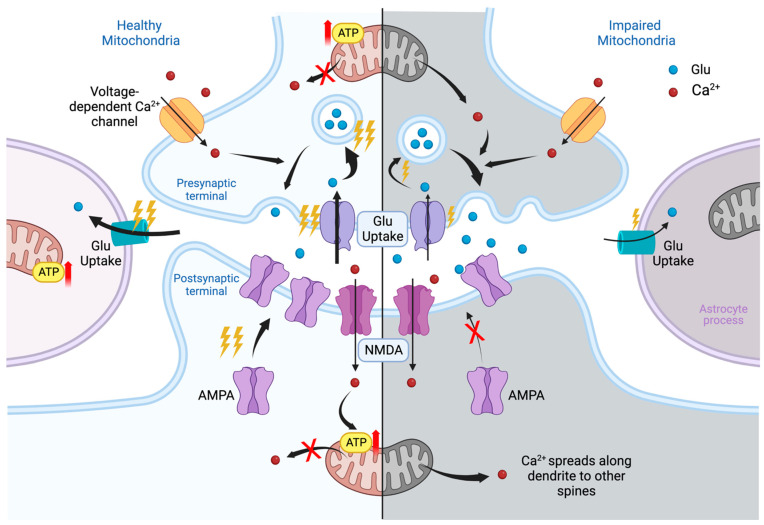
Glutamatergic synapse in healthy (**left**) and impaired (**right**) mitochondria. The synapse consists of three parts: on top—the presynaptic terminal, on the bottom—the postsynaptic region, on the side—a process of astroglia. In the presynaptic terminal, the mitochondrion provides energy for the reuptake of glutamate (Glu), previously released into synaptic cleft, the formation of vesicles containing Glu, and, finally, the release of neurotransmitters into the postsynaptic terminal, as shown in the figure. Ca^2+^ enters the presynaptic terminal through voltage-dependent ion channels; binds to synaptotagmin, a membrane-trafficking protein holding two calcium-binding domains; and thereby participates in the release of Glu. Healthy mitochondria regulate intracellular Ca^2+^. In case of disruption of the mitochondria, the energy supply of all presynaptic processes deteriorates and the levels of neurotransmitters decreases. Nevertheless, dysregulation of Ca^2+^ homeostasis at the initial stage increases the probability of transmitter release, thereby elevating its overall concentration at the synaptic cleft. Astrocytic leaflet provides the reuptake of Glu and regulates perisynaptic potassium levels. Impairment of the mitochondria in astrocyte processes violates reuptake of neurotransmitters. Its excess in synaptic cleft creates conditions for excessive excitation of the postsynaptic membrane and the effect of excitotoxicity. The main events of synaptic plasticity occur at the postsynaptic terminal. The largest and most active dendritic spines of mushroom type are supported by nearby shaft mitochondria clusters. The initial plasticity events are believed to be triggered by Ca^2+^ entry via N-methyl-d-aspartate (NMDA) channels. Elevated intra-spine Ca^2+^ levels activate the buffer protein calmodulin and corresponding kinases. Kinases phosphorylate existing α-amino-3-hydroxy-5-methyl-4-isoxazolepropionic acid (AMPA) receptors; promote the recruitment of extra-synaptic ones, which enhances the signal transduction; and further trigger long-term synaptic modifications. Calcium influx is also required to activate ATP production by local mitochondria. Leakage of calcium from mitochondria may be associated with its spread towards the neighboring spines and decrease in local ATP supply. Ultimately, the lack of energy support will weaken the synapse, reduce its plastic potential and create the conditions for loss of its head volume and mushroom shape, degradation, and/or pruning. Thunder symbol indicates level of consuming of ATP. Red arrow means increase of ATP production.

**Figure 2 toxins-15-00202-f002:**
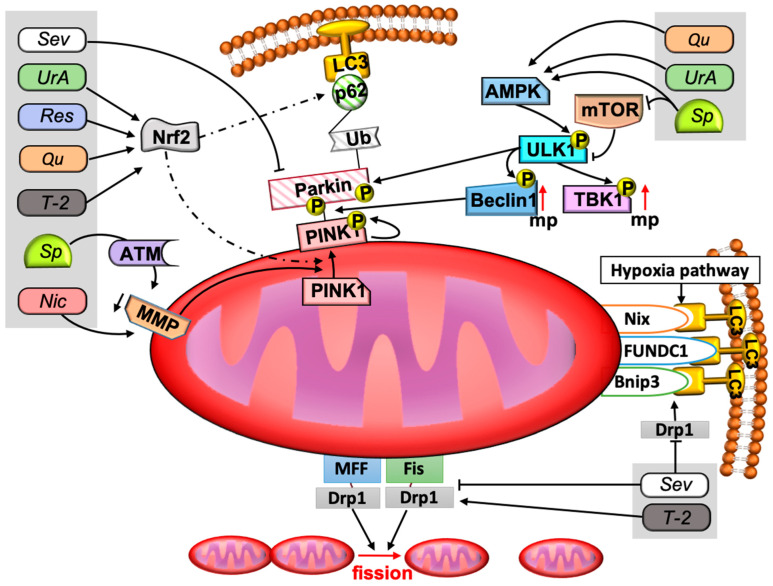
Participation of considered compounds in the signaling pathways of mitophagy. The figure shows the signaling pathways of mitophagy and the place of described compounds in these signaling pathways. Mitophagy is provided in several ways. The main pathway is the Parkin-dependent pathway. A decrease in the membrane potential contributes to the stabilization of PINK1 on the outer mitochondrial membrane. PINK1 phosphorylates Parkin. Parkin is recruited to the outer membrane and provides attachment to the phagophore via ubiquitin chains, p62 protein, and LC3II. Parkin-independent pathways are activated by hypoxia and are provided by Nix, FUNDC1, and Bnip3 molecules. Mitophagy is regulated by AMPK, mTOR, and Nrf2. AMPK and mTOR are an activator and inhibitor of ULK1 kinase, respectively. ULK1 phosphorylates Parkin and other proteins involved in the proper course of mitophagy, such as TBK1 and Beclin1. Nrf2 is a transcription factor that can promote mitophagy under certain circumstances and mitochondrial biogenesis in others. At the lower pole of the mitochondria, the role of Mff, Fis1, and Drp1 has been demonstrated. These factors are very important for proper fission of mitochondria. Drp1 also promotes Bnip3-mediated mitophagy. Resveratrol, urolithin A, quercetin, and T-2 toxin contribute to the regulation of the mitophagy process through the Nrf2 factor. Niclosamide and spermidine reduce the membrane potential of the mitochondria, and spermidine has this effect indirectly, through ATM. Urolithin A, spermidine, and quercetin can also stimulate AMPK, and spermidine blocks mTOR. Sevoflurane reduces p62 and Parkin, and sevoflurane is also able to inhibit mitochondrial division and the Bnip3-mediated mitophagy pathway by reducing Drp1. Abbreviations: AMPK—AMP kinase, ATM—ataxia-telangiectasia mutated protein kinase (ATM), Drp1—Dynamin-related protein 1, LC3—1A/1B-light chain 3 II (LC3II) protein, MMP—mitochondria membrane potential, mTOR—mammalian target of rapamycin, MFF—mitochondrial fission factor, Fis—fission 1, mp—mitophagy, Nic—niclosamide, Nrf2—the transcription factor nuclear factor erythroid 2-related factor 2 (Nrf2), p in yellow circles means phosphorylation of protein, PINK1—PTEN-induced putative kinase 1, Qu—quercetin, Res—resveratrol, Sev—sevoflurane, Sp—spermidine, TBK1—TANK-binding kinase 1, T-2—T-2 toxin, Ub—ubiquitin, ULK1—unc-51-like kinase 1, UrA—urolithin A. Solid lines with arrowheads correspond to activating or promoting effects and inhibiting effects in the case of lines without arrowheads. Dotted line means regulation of expression.

**Figure 3 toxins-15-00202-f003:**
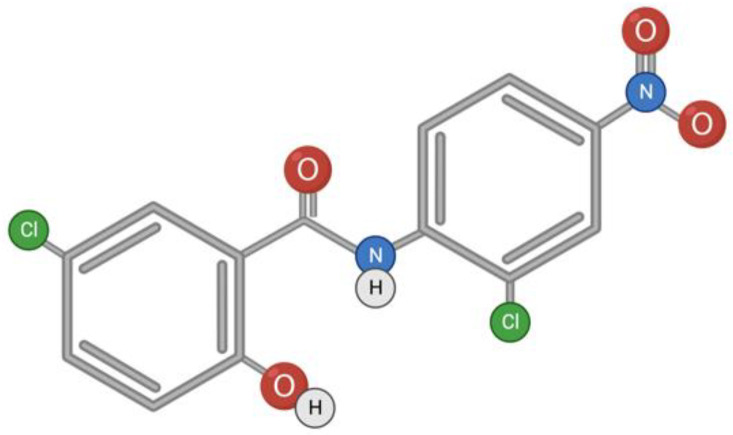
Structural formula of niclosamide.

**Figure 4 toxins-15-00202-f004:**
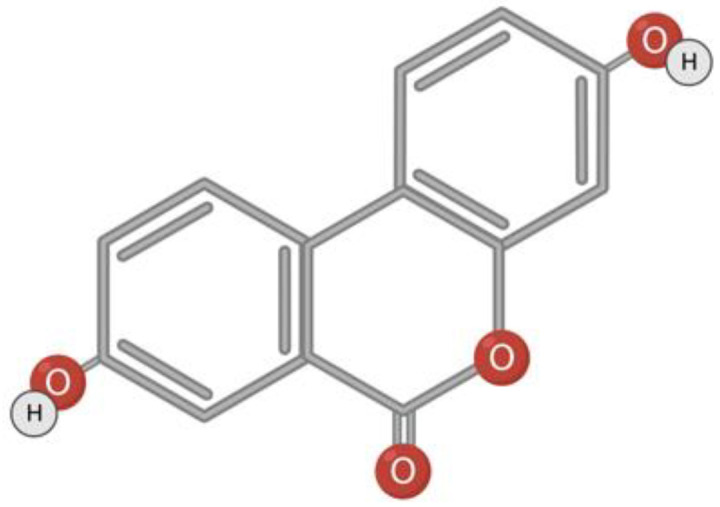
Structural formula of urolithin A.

**Figure 5 toxins-15-00202-f005:**
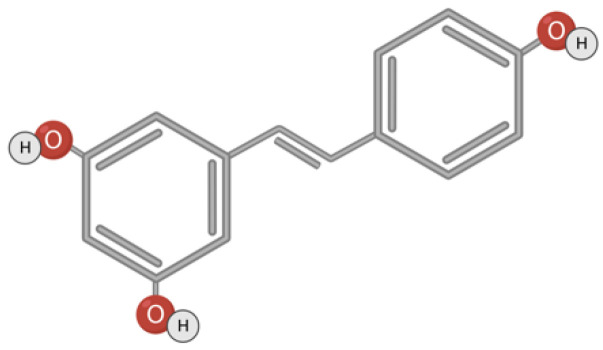
Structural formula of resveratrol.

**Figure 6 toxins-15-00202-f006:**
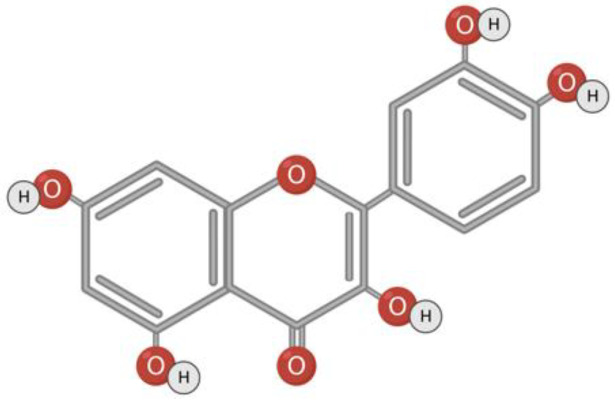
Structural formula of quercetin.

**Figure 7 toxins-15-00202-f007:**
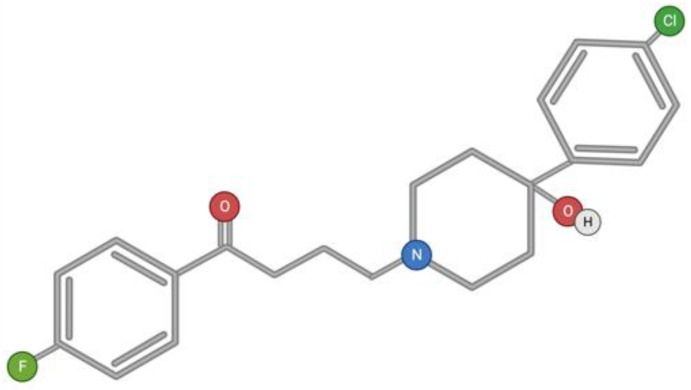
Structural formula of haloperidol.

**Figure 8 toxins-15-00202-f008:**
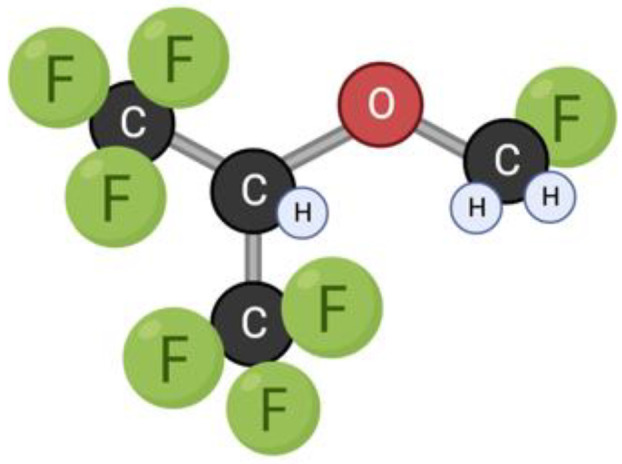
Structural formula of sevoflurane.

**Figure 9 toxins-15-00202-f009:**
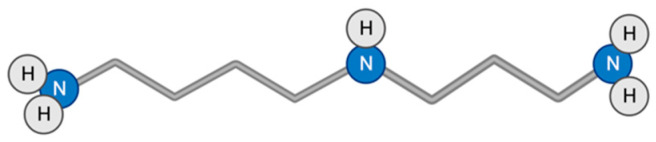
Structural formula of spermidine.

**Figure 10 toxins-15-00202-f010:**
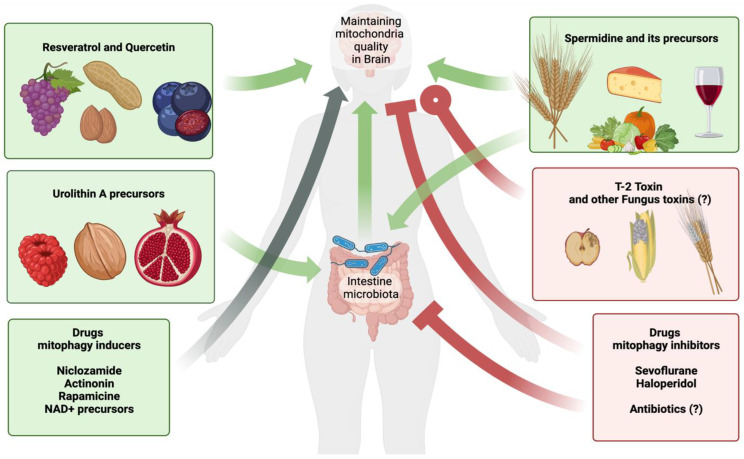
Effect of various compounds on mitophagy and neurodegenerative pathologies. The process of mitophagy is susceptible to external factors, including various substances. These substances are able to directly or indirectly influence through the metabolites of the intestinal microflora. Urolithin A, resveratrol, quercetin, and spermidine are known for their overall positive effects on human health in a wide variety of clinical areas, including cognitive function and memory. Their influence on mitophagy can be considered proven both in vitro and in vivo. In nature, these compounds and their metabolic precursors are found in various plant foods. These products can be recommended for patients at risk of neurodegenerative diseases as part of the diet. Clinical trials are underway to provide evidence of the effectiveness of these drugs in terms of mitophagy induction and neurodegenerative pathologies. Some drugs act as inducers of mitophagy and are suggested to be drugs for the treatment of neurodegenerative pathologies. These drugs include niclosamide, the anthelmintic drug, the antibiotic actinonin, the anticancer drug rapamycin, and NAD+ precursors (last three are not discussed in the text). The effectiveness of these drugs for the treatment of neurodegenerative pathologies by interfering with mitophagy remains to be proven. Sevoflurane and haloperidol have been isolated as inhibitors of mitophagy. Sevoflurane is an ether drug for inhalation anesthesia; its use in medical practice is accompanied by postoperative side effects associated with disruption of the nervous system. It is very likely that these disorders are directly related to the suppression of mitophagy in nerve cells. It has been proven that sevoflurane is able to suppress mitophagy. Haloperidol is an antipsychotic drug and is prescribed for psychiatric patients. Long-term use of haloperidol leads to neurotoxicity. Inhibition of mitophagy also makes some contribution to this neurotoxicity. There are data on the role of intestinal microflora in the process of mitophagy, including in the brain. Antibiotic use is associated with reduced gut microflora function, so we hypothesize that long-term antibiotic use and the resulting gut dysbiosis may lead to disruption of mitophagy in the brain with subsequent effects on neural function. The possible complications associated with the effect on mitochondria should be noted when prescribing any therapy. T-2 toxin is produced by certain types of Fusarium mycotoxins, which can affect various crops. Poisoning with this toxin contributes to neurological symptoms, and at the cellular level, T-2 contributes to mitochondrial damage with subsequent production of ROS. There is evidence that, in certain doses, this toxin is able to stimulate mitophagy, so it may in time become a drug for the treatment of neurodegenerative pathology. We hypothesize that there may be other food-associated fungal and bacterial toxins that can affect mitophagy in brain cells, including for a long time. Notation. Green color of line indicates generally healthy affection on nervous system. The gray—unknown or dosage-depending effect on health. The red means that the substance negatively affects the nervous system in general. Arrows correspond to regulation or stimulation of mitophagy, while rectangles endings suggest the inhibitory influence. Circle end represents unclear affectation on the process of mitophagy. “?” means unexplored or unproven effect.

**Table 1 toxins-15-00202-t001:** Mechanism of action and known effects on the nervous system.

Compound and Its Mode of Action (MoA)	Known Effects on Neural Tissue Cells, Cell Cultures, Animals, and Humans	References (Ref)
**Niclosamide** **(NIC)** activates PINK1 in cells due to reversible damage to the mitochondrial membrane potential.	In ALS cell model NIC attenuates morphological changes, activates mitophagy via the PINK1-Parkin-ubiquitin pathway, prevents TDP-43 mislocalization, and promotes degradation of TDP-43 aggregates;NIC attenuates Paclitaxel-induced thermal hyperalgesia in drosophila flies via PINK1 induction.	
[75]


[24]
MoA ref: [74].
**Toxin T-2 (T-2)** activates Nrf2/PINK1/Parkin pathway;induces PINK1/Parkin-mediated mitophagy;may activate Drp-1-mediated mitophagy leading to apoptosis.	T-2 neurotoxicity caused by ROS, oxidative stress, mitochondrial dysfunction, cell cycle arrest, DNA damage, and inflammatory response;In BV2 microglial cells T-2 is able to induce a loss of mitochondrial membrane potential, increase expression of Beclin1 and LC3II proteins, and to activate autophagy, protecting cells;Neurological symptoms of poisoning: muscle weakness, depression and ataxia, and anorexia.	
[76,77]


[78]


[79]
MoA ref: [80,81,82].
**Urolithin A (UrA)** activates Nrf2 regularizing mitophagy and mitochondrial biogenesis;activates AMPK pathway.	In AD models of SH-SY5Y cells and in vivo, UrA prevented destruction of the AIP-AhR complex and suppressed expression of APP and BACE1;In microglia, UrA alleviates neuroinflammation and enhances phagocytosis by enhancing mitophagy, leading to a decrease in the accumulation of tau-protein and beta-amyloid;In BV2 microglial cell PD model, UrA, due to mitophagy, reduced neuroinflammation and provided a neuroprotective effect;UrA attenuates memory impairment and neuroinflammation in APP/PS1 mice;UrA appeared to be safe and induced the molecular signature of improved mitochondrial and cellular health in the elderly.	
[44]


[43,83]


[84]


[85]

[86]
MoA ref: [66,67,68,69,85,87]
**Resveratrol** **(RES)** reduces oxidative stress by suppressing the formation of ROS, derivatives of NADPH oxidase;activates Nrf2 regularizing mitophagy and mitochondrial biogenesis;in endothelial dysfunction, resveratrol activated mitophagy through Bnip3.	In a model of AD cells, RES promoted mitophagy, attenuating oxidative damage to neurons caused by Aβ-peptide;In a cadmium-induced mitophagy model, RES attenuated the overexpression of p62 and PINK1/Parkin, which suppressed mitophagy and ultimately restored mitochondrial homeostasis	
[88]


[89]

MoA ref: [89,90,91]
**Quercetin (QU)** activates AMPK and Nrf2, contributing to the modulation of the expression and activity of PINK1/Parkin-mediated mitophagy.	QU decreased levels of hyperphosphorylated Tau and amyloid plaques in AD models, leveled out neuroinflammation, inhibited microglia and astrocyte activation, and provided a mitoprotective effect, an increase in ATP production, and a decrease in ROS levels;In PD rat models, QU reduced oxidative stress, increased PINK1 and Parkin, and reduced α-synuclein expression in vitro, and in vivo, QU mitigated progressive motor disorders, reduced neuronal death, and reduced mitochondrial damage and α-synuclein accumulation;In microglial cells, QU stimulates mitophagy, preventing neuronal damage by inhibiting mtROS-mediated activation of the NLRP3 inflammasome.	

[92,93]



[94]



[95]
MoA ref: [96,97]
**Haloperidol** **(HAL)**	In long-term use, HAL causes oxidative-stress-induced neurotoxic effects with cognitive dysfunction and tardive dyskinesia;Reduced the level of autophagy in SH-SY5Y cells and rat neurons;In in vivo MS models, HAL inhibited autophagy and mitophagy, restored myelin production and axonal myelination, and contributed to the improvement of motor activity in mice.	[25,98,99,100]

[25,26]

[49]
**Sevoflurane** **(SEV)** reduces excessive production of Drp1 and Parkin.	SEV associated with postoperative complications with various cognitive impairments;SEV stimulates the apoptosis pathway, providing a neurodegenerative effect in hippocampal neuronal HT22;mitophagy inducers contributed to a decrease in the toxic effect of SEV;SEV attenuates brain damage inhibiting autophagy and apoptosis in cerebral ischemia reperfusion rats	
[21,101,102]



[20,21]

[21,22,103]

[23]
MoA ref: [104,105]
**Spermidine** **(SP)** activates ATM, induces mitochondrial membrane depolarization promoting the PINK1/Parkin-dependent mitophagy pathway;inhibits mTOR;activates AMPK.	SP improved olfactory memory in models with age-related memory impairment drosophila flies;In C. elegans models of AD and PD, SP inhibited memory loss in AD models and improved behavioral performance in PD models;In accelerated aging SAMP8 model of mice, SP increased the expression of neurotrophic factors in neurons and reduced memory loss in the ORT and the open field test OFT;In older adults, SP improved episodic memory in MST.	
[106]


[107]

[108]

[109,110]
MoA ref: [54,107,111,112,113]

## Data Availability

Not applicable.

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
