# Peer review of "Differential Role of Active Compounds in Mitophagy and Related Neurodegenerative Diseases"

_toxins, 2023, doi:10.3390/toxins15030202_

Round 1
Reviewer 1 Report
This review article discussed the role of toxins in mitophagy and neurodegeneration. The review is very interesting and highlight an area that is not commonly talk about in the field of neurodegenerative disorders. I have the following major concerns that the authors may wish to consider
1. The reference in the text does not match patterns in the reference list. This makes it difficult to check references. Can the authors change the text citations format to match does in the reference list?
2. Provide a table that summarized all the toxins discussed, their mode of action, potential target, disease/neurodegenerative disorder and reference. This will help the readers to follow
3. Amyloid beta is associated with plaques not fibrils can the authors double check this statement
4. Can the authors provide examples of altered mitochondria synapse that results in disorders rather than saying wide range of disorders?
5. Please check “trough” I think it is “through” lines 114
6. What is EAAT-1 and 2?
Reviewer 2 Report
The idea (title) of this review article is interesting. Although the title is “Role of toxins~”, the compounds described in this article are not toxins except Toxin T-2.
The word “neurodegeneration” means loss of structure or function of neurons. In this paper, authors described the mitophagy of both neuron and glial cells. So that, the title does not reflect the content.
First of all, this paper is difficult to read because of the bad format. 1, Some part is not clear whether it is main text or figure legend (for example page13~14). 2, Figure legends do not explain the figure. 3, Although references are listed, author name and year are written in the main text. 4, Some abbreviation are not spelled-out.
Additionally, English is not good. Authors need to have it corrected by a native in English proofreading.
Authors picked up 7 compounds and described each of them. Since it is difficult to understand the whole picture, create a table of summarize maybe good..
Certainly too much or too little mitophagy is not good, but the description of each compound conveniently describes inhibition or activation of mitophagy, which makes the main idea inconsistent and difficult to understand.
Minor comments
Figure 1, need to explain the mean of each symbols and arrows.
Figure 2, shape of proteins and compounds are similar. The shape and/or font should be separated for easy identification.
Figure 3, three blue round item in Resveratrol is unknown.
Bacterial name should be written in italic (page 12)
First letters of some words are capitalized. For example, Line 37, Adenosine Triphosphate ATP => adenosine triphosphate (ATP)
Line 541~542, Niclosamine, Actinonin, Rapamycin
LIne 326, need to explain the cell type of SH-Sy5Y.
Line 58 “This” means what?
Line 114 figure => figure 1
Reviewer 3 Report
This is an interesting review article that is acceptablre after a minor revision. Only view minor points should be adressed:
1) Structural formulas and a short description of chemical properties of the drugs should be added.
2) Figures 1-3 should have more relations with the main text.
3) Since the Authors discussed mitoprotective properties of resveratrol, they should also consider the influence of plant-derived flavonoids, which may also exert mitoprotective actions, such as quercetin, for example.
Round 2
Reviewer 1 Report
The authors have addressed all my concerns.
Author Response
Dear reviewer,
let us thank you very much for your constructive and useful responses, which helped to improve our manuscript and make it better.
Sincerely,
Authors
Reviewer 2 Report
At the first review, I commented that most compounds are not "toxin", so that authors changed the title. However, in the text, they still use toxin several times (line 127, 239, 507 and more). It does not appear that the authors understood the meaning of my comment and changed the title.
Creating Table 1 is good for easier understanding, but the content should be more compact.
The entire text is full of errors and clunky. I think it should be proofread properly. Below I list the mistakes I have noticed, but there must be more mistakes.
Figure 1
Health should be Healthy
What does thunder symbol mean?
Spell out NMDA, AMPA (AMPA is typed APMA in the legend)
line 145, is COX really cyclooxygenase? I'm guessing it's cytochrome c oxidase.
line 181 "on the other hand" repeat twice
line 221, Nrf2 first appear but line 234 it spelled out.
line 280, GH3 is what type of cell?
Author Response
Dear Reviewer,
Thank You for Your contribution. We paid attention to Your valuable comments and made related changes. New changes are marked by yellow.
At the first review, I commented that most compounds are not "toxin", so that authors changed the title. However, in the text, they still use toxin several times (line 127, 239, 507 and more). It does not appear that the authors understood the meaning of my comment and changed the title.
For this round we found all “toxin” words and changed them if it was necessary.
Creating Table 1 is good for easier understanding, but the content should be more compact.
Table 1 is now more compact. We hope it is compact enough.
The entire text is full of errors and clunky. I think it should be proofread properly. Below I list the mistakes I have noticed, but there must be more mistakes.
We managed to fix text with help of native English speaker.
Figure 1
Health should be Healthy
Fixed
What does thunder symbol mean?
We added information in description of figure
Spell out NMDA, AMPA (AMPA is typed APMA in the legend)
NMDA, AMPA are spelled out.
line 145, is COX really cyclooxygenase? I'm guessing it's cytochrome c oxidase.
Yes. This is cytochrome c oxidase. Fixed.
line 181 "on the other hand" repeat twice
line 221, Nrf2 first appear but line 234 it spelled out.
Thank You. It’s fixed.
line 280, GH3 is what type of cell?
This is a rat pituitary cell. Information is added.
Kind regards,
Authors
